# Dual-Layer Anomalous Hall Effect Sensor for Enhanced Accuracy and Range in Magnetic Field Detection

**DOI:** 10.3390/nano15070527

**Published:** 2025-03-31

**Authors:** Sitong An, Lvkang Shen, Tianyu Liu, Yan Wang, Qiuyang Han, Ming Liu

**Affiliations:** School of Microelectronics, Xi’an Jiaotong University, Xi’an 710049, China; ansitong18@stu.xjtu.edu.cn (S.A.);

**Keywords:** anomalous Hall effect, heterostructure, magnetic sensor, multi-range devices

## Abstract

This study introduces a method aimed at enhancing both the accuracy and the range of magnetic field sensors, which are two critical parameters, in a novel NiCo_2_O_4_-based anomalous Hall effect sensor. To fine-tune the linear range of the sensor, we introduced epitaxial strain using a MgAl_2_O_4_ cover layer, which significantly influenced the strain-modulated magnetic anisotropy. A NiCo_2_O_4_/MgAl_2_O_4_/NiCo_2_O_4_/MgAl_2_O_4_ heterostructure was further constructed, achieving differentiation in the material characteristics across both upper and lower NiCo_2_O_4_ layers through the modulation of thickness and strain. A dual-layer Hall bar was designed to enhance the integration of the sensor, offering varied detection ranges. This approach enabled the realization of ultrahigh sensitivity, measuring 10,000 V/(AT) within a ±0.1 mT range, and a competitive sensitivity of 60 V/(AT) within a ±5 mT range. By reducing the thickness of the top NiCo_2_O_4_ layer, an ultra-wide measurement range of ±1000 mT was also achieved. These results highlight the considerable promise of NiCo_2_O_4_-based anomalous Hall effect devices as compact, multi-range tools in the domain of magnetic sensing technology.

## 1. Introduction

The demand for compact, multifunctional magnetic field sensors has surged, driven by their broad applications in areas such as the Internet, smart devices, and power grids [1]. Thin-film devices leveraging phenomena like the anomalous Hall effect (AHE), anisotropic magnetoresistance (AMR), giant magnetoresistance (GMR), and tunnel magnetoresistance (TMR) are prevalent in the fabrication of commercial magnetic sensors [2,3,4,5]. To achieve integrated multi-range detection, a vertical structure is necessary. However, magnetoresistance sensors, characterized by their complex structures and multiple annealing temperatures, present significant challenges to such integration [6]. In contrast, the AHE sensor features a more straightforward structure, thereby simplifying the design and manufacturing processes.

Figure 1a is a schematic diagram of the AHE of NiCo_2_O_4_ (NCO) films utilized in detection applications with different sensitivity. The sensitivity and range are constrained by the intrinsic properties of the thin film, such as the coercive field and Hall resistance [7,8]. The main approach to control magnetic anisotropy in epitaxially grown NCO thin films is by adjusting the growth parameters such as the deposition temperature and the thickness of the film pattern. These adjustments can influence the magnetic anisotropy energy, thereby affecting the coercive field and Hall magnetoresistance, as demonstrated by prior studies [8]. The core of these manipulation techniques lies in controlling the film lattice constants to alter internal stress conditions, subsequently adjusting the perpendicular magnetic anisotropy energy of NCO films. As shown in Figure 1b, modulating the stress conditions of the film by applying additional strain and adjusting the film thickness is a conventional method for tuning the measurement range of magnetic sensing films. For example, the compressive strain induced by epitaxial mismatch reduces the in-plane lattice constant (a < a_0_) and increases the out-of-plane constant (c > c_0_), enhancing sensitivity via strain-driven magnetic anisotropy. Increasing the film thickness relaxes the strain, gradually restoring the lattice constants toward bulk values, which broadens the measurement range at the cost of reduced sensitivity. The letters “a” and “c” in Figure 1b represent the in-plane and the out-of-plane lattice constants, respectively. The subscripts indicate distinct values. “Thickness” refers to the thickness of the epitaxial film.

Research has indicated that the detection range of NCO thin film sensors can be fine-tuned by managing the input current, which underscores the potential in expanding the detection capabilities of single-structure AHE sensors [9]. However, it is challenging to greatly expand the detection range within a specific sensitivity range due to their interplay. Thus, achieving the desired detection range and accuracy simultaneously from a single-structure AHE sensor is difficult. This investigation demonstrates that the application of additional stress through the strategic growth coverage of identical MgAl_2_O_4_ (MAO) film patterns enabled further refinement of the coercive force and Hall resistance in NCO films. A MAO(10)/NCO(5)/MAO heterostructure was engineered, where the figures in parentheses represent the thickness of each layer in nanometers, achieving the targeted sensitivity and linear range for magnetic field detection by modulating the input current. Building on this foundation, a subsequent layer of NCO film was deposited atop the existing pattern. This layered approach resulted in a sensor capable of ultrahigh sensitivity, registering 10,000 V/(AT), and an ultra-wide measurement range of ±1000 mT within distinct layers of the same sensor. This advancement seamlessly aligns with the previous discussion on the simplicity of structure and the methodical control over magnetic anisotropy through interlayer stress, highlighting a coherent progression towards the realization of sensors with both enhanced sensitivity and an expanded detection range in the domain of magnetic field sensing.

## 2. Materials and Methods

NCO thin films along the (001) direction were epitaxially grown on (001) MAO substrates by pulsed laser deposition. The thickness of the films was 5 nm [10]. The growth temperature was maintained at 350 °C, and the oxygen pressure was set to 130 mTorr. Subsequently, additional (001) MAO thin films with the thickness of 5 nm, 10 nm, and 20 nm were epitaxially grown on the former samples by pulsed laser deposition, separately. The growth temperature was maintained at 350 °C, and the oxygen pressure was set to 65 mTorr. The crystal structures of four heterostructures were characterized by X-ray diffraction (XRD), separately. Hall bar patterns were fabricated on these heterostructures by the standard lithography process. The length of the Hall bars was 400 μm, while the width was around 100 μm. Figure 1c is a schematic diagram of a single-structure AHE sensor. Finally, two NCO Hall bar patterns of different thicknesses were grown on the previous Hall bar patterns, which were made from the MAO(10)/NCO(5)/MAO heterostructure. The growth temperature was maintained at 350 °C, and the oxygen pressure was set to 130 mTorr. The length of the Hall bars was 400 μm, while the width was around 90 μm.

## 3. Results

We observed that the coercive field of the AHE in the NCO films could be further regulated by growing the MAO films shown in Figure 2a. The hysteresis loop had a nearly-zero coercive field after growing the 10 nm or 20 nm MAO films on the top of the former films, which meets the requirements of magnetic sensors. Figure 2c shows that the coercive field of the heterostructure rapidly decreased with an increasing MAO thickness. The reduction rate stabilized after the MAO films reached 10 nm. The statistical results for the saturated Hall resistance and longitudinal resistance of each heterostructure are presented in Figure 2d. We observed that the saturated Hall resistance was positively correlated with the longitudinal resistance at room temperature, which is consistent with previous studies [11]. Reciprocal space mapping (RSM) techniques were employed to analyze the stress state of the NCO films, as illustrated in Figure 2b. We observed that in all heterostructures, the NCO films were fully epitaxially grown on the MAO substrates, with the in-plane lattice constants fixed at 8.08 Å due to lattice matching. The out-of-plane lattice constants (c-axis) of the NCO films, however, varied with the thickness of the overlying MAO layer. When the MAO thickness increased from 0 nm to 20 nm, the c-axis decreased from 8.18 Å to 8.16 Å, approaching the bulk lattice constant of NCO (8.15 Å). This reduction in the c-axis indicated a gradual relaxation of the tensile strain along the out-of-plane direction as the MAO layer thickened. The compressive in-plane stress from the MAO substrate induced perpendicular magnetic anisotropy (PMA) in the NCO films [12]. The decrease in the c-axis with thicker MAO layers weakened this PMA, which directly correlated with the observed reduction in the coercive field (Figure 2c). Additionally, the longitudinal resistance of the NCO films exhibited a non-monotonic trend (Figure 2d). Initially, the resistance increased due to reduced oxygen vacancies in NCO during high-oxygen-pressure MAO growth, altering the Ni valence state. As the MAO thickness further increased, strain relaxation dominated, diminishing the spin–orbit scattering and lowering the resistance. This interplay between oxygen vacancy modulation and strain relaxation explains the resistance behavior [13].

As previously shown, oxygen vacancies have a significant effect on the AHE, and we performed in situ annealing of these four films with different structures at half an atmosphere of high oxygen pressure and 350 °C. The AHEs were subsequently measured and comparatively analyzed, as shown in Figure 3. In the Figure, the upper portion indicates the results before annealing, and the lower portion those after annealing. It can be seen that, except for the film not covered with the MAO film, the coercive field after annealing increased to different degrees, and the Hall resistance value was slightly reduced. In addition, as the thickness of the MAO film increased, the increase in coercive field was gradually reduced. We believe that the decrease in Hall resistance and the increase in coercive field were caused by a decrease in oxygen vacancies, and the upper layer of MAO had a certain inhibiting effect on this change. However, the NCO films not covered with the MAO film lacked this inhibition, which led to a drastic change in the Hall resistance value. So, it was hard to further tune the sensitivity by annealing on the basis of the applied stress.

The current can modulate sensitivity as well. Figure 4a illustrates the relationship between Hall resistance and magnetic field for different heterostructures at varying input currents ranging from 0.1 mA to 2.0 mA. As the current density increased, we observed that the Hall resistance curve transitioned from a highly rectangular shape, which indicated a significant non-zero coercive field, to a more linear shape approaching a nearly-zero coercive field. We conducted statistical analyses of the coercive force variation for each heterostructure under different currents, as shown in Figure 4b. It was evident that the coercive force rapidly decreased with the increasing current of the first two heterostructures. Conversely, for the last two heterostructures, the coercive force was already close to zero at low currents, resulting in a smaller reduction rate as the current increased. Figure 4c depicts the relationship between saturated Hall resistance and input current, where saturated Hall resistance diminishes with increasing current, which is a phenomenon exploitable in designing dynamically adjustable sensing systems. The change in Hall resistance is primarily induced by Joule heating due to the current density. This heating disrupts the magnetic equilibrium and weakens the out-of-plane magnetization strength [14]. In a comprehensive comparison, we leaned towards selecting the MAO(10)/NCO(5)/MAO heterostructure to construct an AHE sensor that would exhibit higher saturated Hall resistance and nearly-zero coercive force at various currents. By calculation using the experimental slope (dR/dH) of the Hall resistance signal within the ±0.1 mT linear range, an ultrahigh sensitivity of 10,000 V/(A·T) under a 1 mA input current was achieved. This greatly will improve the detection accuracy of weak magnetic field signals.

## 4. Discussion

We developed a dual-layer anomalous Hall effect (AHE) sensor with dynamically adjustable detection ranges. The sensor integrates two vertically stacked Hall bar structures, i.e., a lower layer with a 5 nm thick NCO film and an upper layer with a variable-thickness NCO film, separated by a 10 nm MgAl_2_O_4_ (MAO) insulating spacer (structural details in Figure 5a. The lower NCO layer and the insulating spacer together constitute the bottom layer). Figure 5c illustrates the Hall signal of the top layer with a 7.5 nm thick NCO film. In addition to detecting the weak magnetic field in the original lower layer, a detection range of ±5 mT and sensitivity of 60 V/(AT) were achieved by connecting the upper-layer Hall bar pattern under a 6 mA input current. To address the critical demand for wide-range magnetic field monitoring in power grid applications, we developed another dual-layer anomalous Hall effect (AHE) sensor with an upper layer covered with a 6 nm thick NCO film. The lower layer, measured under a 1 mA input current, achieved a detection range of ±0.1 mT with a sensitivity of 10,000 V/(A·T) (Figure 5b), enabling the precise monitoring of weak magnetic fields (e.g., ~0.1 mT generated by 50 A DC currents). In contrast, the upper layer, operated at 1 mA, exhibited a significantly broader detection range of ±1000 mT with a reduced sensitivity of 0.3 V/(A·T) (Figure 5d). This abrupt change in performance was attributed to the reduced NCO thickness in the upper layer, which altered the magnetic anisotropy due to increased surface roughness at the NCO/MAO interface.

The sensor connections are shown in Figure 6a. To ensure that the magnetic field was perpendicular to the sensor surface, we placed the sensor inside a concentrator. The operational logic of the dual-layer sensor is illustrated in Figure 6b. During normal circuit operation, the weak magnetic field (~0.1 mT) generated by the power line falls within the detection range of both layers. The lower layer’s higher sensitivity ensures accurate current monitoring. When a short circuit occurs, the resulting surge in current generates a magnetic field exceeding the lower layer’s ±0.1 mT range. However, the upper layer’s ultra-wide range (±1000 mT) remains functional, triggering a short-circuit alarm. The error conditions were defined as follows: if the upper layer exceeds its range while the lower layer operates normally, a sensor anomaly is flagged; and if both layers exceed their ranges, this indicates an extreme fault (e.g., catastrophic overcurrent or sensor damage), requiring on-site inspection.

The dual-layer AHE sensor architecture developed in this study directly addresses the critical challenge of simultaneously enhancing measurement accuracy and detection range in magnetic field sensing. By integrating vertically stacked Hall bar structures with distinct NCO layer thicknesses (a 5 nm lower layer and a 6 nm upper layer) separated by a 10 nm MAO insulating spacer (Figure 5a), we achieved complementary performance characteristics that surpassed those of conventional single-layer sensors. For weak magnetic fields such as the 0.1 mT signal generated by 1~50 A grid currents, the lower layer provided a sensitivity of 10,000 V/(A·T) within a ±0.1 mT range, enabling high-precision monitoring. In contrast, the upper layer extended the detection capability to ±1000 mT with a reduced sensitivity of 0.3 V/(A·T), ensuring reliable operation during short-circuit events or extreme over currents. This dual-range functionality eliminates the traditional trade-off between sensitivity and range inherent to single-structure AHE sensors. This architecture enables the multifunctional monitoring of normal currents, short circuits, and sensor anomalies, leveraging the complementary ranges and sensitivities of the dual-layer design. The MAO spacer’s electrical insulation ensures the independent operation of the two layers, while the thickness-modulated NCO films provide distinct magnetic responses. This architecture uniquely combines precision and breadth, fulfilling the title’s promise of enhanced accuracy and range in magnetic field detection.

## 5. Conclusions

In summary, we successfully developed a single-structure AHE sensor based on a MAO/NCO/MAO heterostructure, which demonstrated minimal coercive force while maintaining an ultrahigh sensitivity of 10,000 V/(AT). Furthermore, we established a dual-layer AHE sensor using an NCO/MAO/NCO/MAO heterostructure as a base. By adjusting the thickness of the top layer, a dual-layer AHE sensor with one layer acting as a sensor for small magnetic fields in the range ±5 mT and the other for large fields of ±1000 mT, was realized. These findings suggest that the devices employing NCO as a foundation possess significant potential for integrated diverse applications.

## Figures and Tables

**Figure 1 nanomaterials-15-00527-f001:**
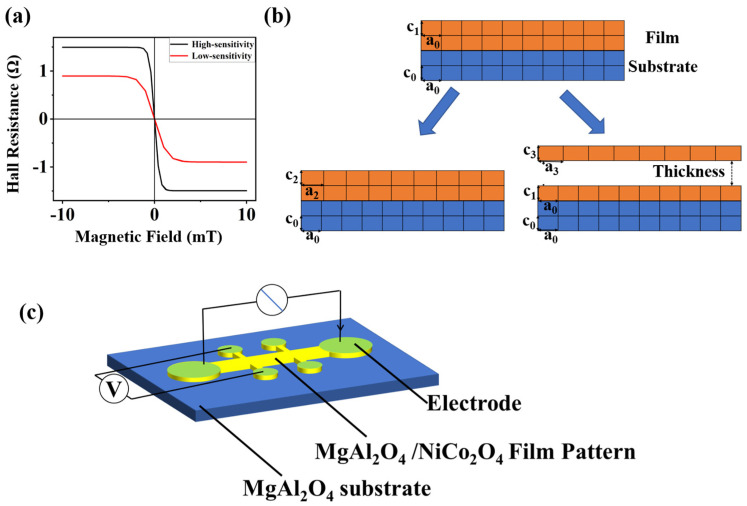
(**a**) Schematic diagram illustrating different Hall resistance signals from high-sensitivity profiles and low-sensitivity profiles. (**b**) The main methods of regulating the stress condition of the film. (**c**) Schematic diagram of the single-structure AHE sensor.

**Figure 2 nanomaterials-15-00527-f002:**
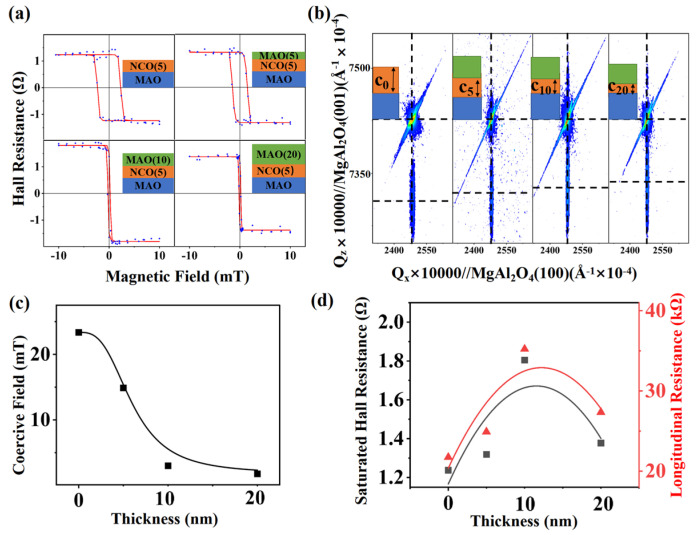
(**a**) Hall resistance vs. magnetic field hysteresis of four kinds of heterostructures at room temperature under a 0.5 mA input current. (**b**) Reciprocal space mapping taken around the NCO(206) films in four kinds of heterostructures. The insets show the lattice constant variation (c is the out-of-plane lattice constant. The value in the bottom right corner refers to the thickness of the MAO films). (**c**) The thickness of the MAO films influences the coercive field of the Hall bar. (**d**) The thickness of the MAO films influences the saturated Hall resistance and the longitudinal resistance of the Hall bar.

**Figure 3 nanomaterials-15-00527-f003:**
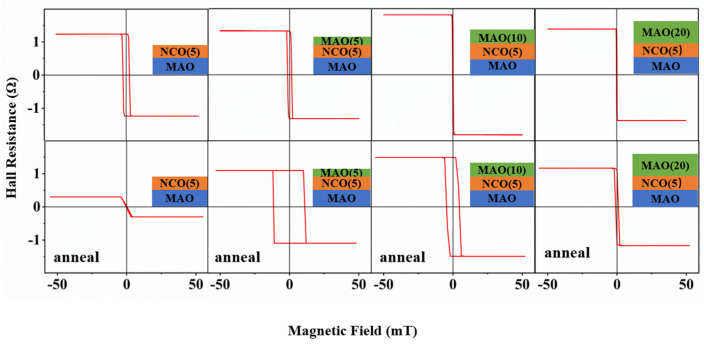
Hall resistance before and after annealing of four heterostructures vs. magnetic field hysteresis at room temperature under a 0.5 mA input current.

**Figure 4 nanomaterials-15-00527-f004:**
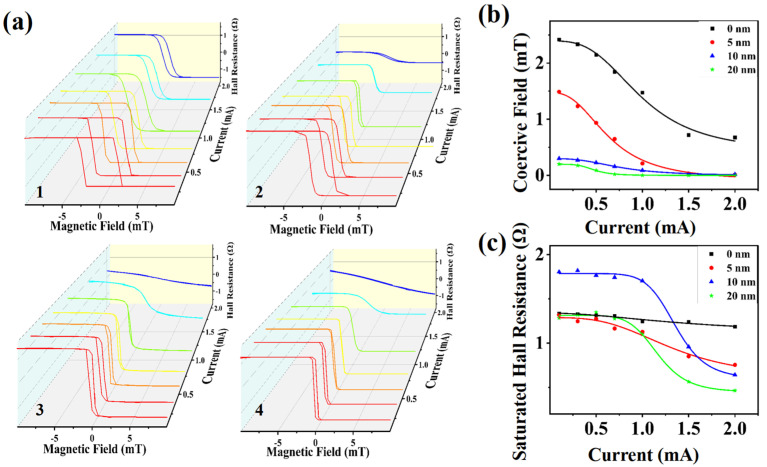
(**a**) Current-dependent Hall resistance vs. magnetic field of the four kinds of heterostructures with a different input current in the range of 0.1–2.0 mA. Illustrations 1–4 refer to NCO(5)/MAO, MAO(5)/NCO(5)/MAO, MAO(10)/NCO(5)/MAO, and MAO(20)/NCO(5)/MAO, respectively. (**b**) Current-dependent coercive field of the four kinds of heterostructures. (**c**) Current-dependent saturated Hall resistance of the four kinds of heterostructures. The value in the top right corner refers to the thickness of the MAO films.

**Figure 5 nanomaterials-15-00527-f005:**
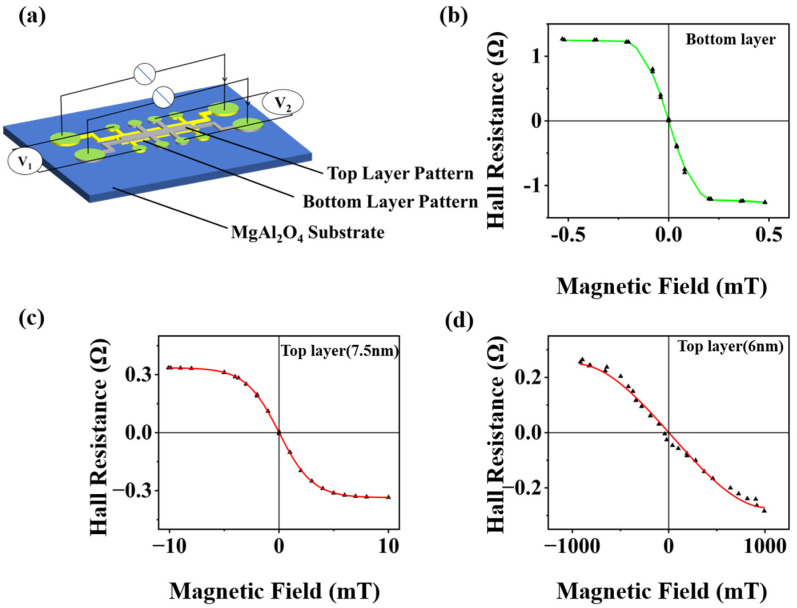
(**a**) Schematic diagram of the dual-layer AHE sensor. (**b**) Hall resistance vs. magnetic field hysteresis of bottom Hall bar at room temperature under a 1 mA input current. (**c**) Hall resistance vs. magnetic field hysteresis of top Hall bar at room temperature under a 6 mA input current. (**d**) Hall resistance vs. magnetic field hysteresis of another top Hall bar at room temperature under a 1 mA input current.

**Figure 6 nanomaterials-15-00527-f006:**
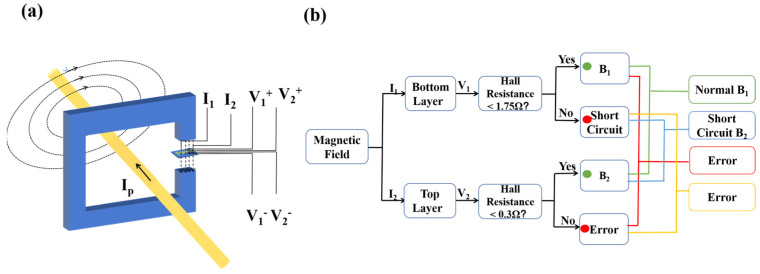
(**a**) Schematic diagram of the sensor connections. (**b**) The output logic of a dual-layer AHE sensor with multifunctional monitoring.

## Data Availability

All data generated or analyzed during this study are included in this study.

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
