# Peer review of "Dual-Layer Anomalous Hall Effect Sensor for Enhanced Accuracy and Range in Magnetic Field Detection"

_nanomaterials, 2025, doi:10.3390/nano15070527_

Round 1

Reviewer 1 Report

Comments and Suggestions for Authors

The paper describes design, fabrication and characterization of a dual-layer anomalous Hall effect sensor. By the introduction of epitaxial strain in the heterostructure, the magnetic anisotropy could influenced, leading to an adjustable magnetic field sensing range. A dual-layer Hall sensor was realized, with one layer for sensitive detection of small magnetic fields, and a second layer for measurement of strong magnetic field. I think this is a truly useful improvement for Hall sensors. To the best of my knowledge, the work is novel and original. The title of the paper is appropriate. The abstract summarizes well the approach and states the most important results. The paper is well structured. In the introduction, the work is put into proper context to previously published research. I think that Materials and Methods, Results and Discussion are mostly well elaborated. I only missed an in-depth explanation of the most important result, the dual-layer, dual-range Hall sensor.

In the conclusion, the main findings are briefly summarized. In revision, I suggest to address the following issues: 1) Lines 75 and 82: The oxygen pressures should be given in its SI unit, Pascal (Pa). mT (milli-Tesla) is a magnetic field unit. Or do you mean milli-Torr? But that’s an outdated unit, and it is not abbreviated with T.

2) Lines 157-158: I don’t understand the sentence “By calculation, an ultrahigh sensitivity 157 of 10000 V/(AT) in the range of ±0.1 mT under 1 mA input current were realized.” Is it a theoretical value or an experimentally obtained value? How did you determine it? Please explain that in detail because it is a very important result that you emphasize already in the abstract.

3) Lines 181-185: Your presentation of the dual-layer Hall sensor with the two ranges for magnetic field measurement is very short and somewhat superficial. I missed an in-depth explanation of this very important result. In particular, I missed an explanation of the schematic logic diagram in Figure 5(d). I suggest to add a detailed explanation of this diagram, explaining how the range switching is performed, what is meant by “short circuit” here (why should a resistance below 1.75 Ohms be a real short circuit?), and under which conditions the two error cases will be reached. I suggest to elaborate comprehensively on this novel structure, and to also provide more details in the caption of Figure 5. Figures 5(a) and (b) should be enhanced in size, or an inset showing the details around zero magnetic field should be added. From your current figure, I found it almost impossible to estimate the slope of the bottom layer.

4) Line 197: “Two different ranges … contains ±5 mT … can be achieved” is inappropriate wording. I suggest to rewrite it in the following fashion: “A sensor with two layers, one acting as a sensor for small magnetic fields in the range ±5 mT and one for large fields of ±1000 mT, was realized.” That’s just an idea, please modify as you like. In addition, I found the following typos:

5) Line 124: correctly it should read: “As previously studied, ..”

6) Figure 5(d): “Magnetic field” in the left box.

Author Response

Comments 1: Lines 75 and 82: The oxygen pressures should be given in its SI unit, Pascal (Pa). mT (milli-Tesla) is a magnetic field unit. Or do you mean milli-Torr? But that’s an outdated unit, and it is not abbreviated with T.

Response 1: Thank you for your kind reminder. We sincerely apologize for the unit error. The oxygen pressures in Lines 75 and 82 should indeed be given in ​mTorr​ (milli-Torr, 1 mTorr = 0.133 Pa). The abbreviation "mT" was a typographical mistake, and we have corrected it to ​mTorr​ in the revised manuscript. We appreciate your careful attention to this detail.This revision ensures clarity and adherence to standard unit conventions.

Comments 2: Lines 157-158: I don’t understand the sentence “By calculation, an ultrahigh sensitivity 157 of 10000 V/(AT) in the range of ±0.1 mT under 1 mA input current were realized.” Is it a theoretical value or an experimentally obtained value? How did you determine it? Please explain that in detail because it is a very important result that you emphasize already in the abstract.

Response 2: Thank you for raising this critical question, and I sincerely apologize for the lack of clarity in our original description. The sensitivity value of 10,000 V/(A·T) ​is an experimentally derived result, not a theoretical prediction. Here’s how we determined it:

The sensitivity was calculated directly from the ​experimentally measured Hall resistance signal​ (R) versus applied magnetic field (H). Specifically: The ​slope of the linear region​ (dR/dH) in the R-H curve was extracted from experimental data. The linear range​ (±0.1 mT) was also determined experimentally by observing the deviation from linearity in the R-H curve. The sensitivity (S) was calculated using the standard definition for Hall sensors: S=dR/dH. Here, ”dR” is the change in Hall resistance, and “dH” is the corresponding change in magnetic field within the linear range. The sensitivity was normalized to the input current (1 mA) to ensure consistency with conventional Hall sensor metrics (units: V/(A·T)).We recognize the ambiguity in the original wording and will revise the text to explicitly state:
“By calculation using the experimental slope (dR/dH) of the Hall resistance signal within the ±0.1 mT linear range, an ultrahigh sensitivity of 10,000 V/(A·T) under 1 mA input current was achieved.”

Comments 3: Lines 181-185: Your presentation of the dual-layer Hall sensor with the two ranges for magnetic field measurement is very short and somewhat superficial. I missed an in-depth explanation of this very important result. In particular, I missed an explanation of the schematic logic diagram in Figure 5(d). I suggest to add a detailed explanation of this diagram, explaining how the range switching is performed, what is meant by “short circuit” here (why should a resistance below 1.75 Ohms be a real short circuit?), and under which conditions the two error cases will be reached. I suggest to elaborate comprehensively on this novel structure, and to also provide more details in the caption of Figure 5. Figures 5(a) and (b) should be enhanced in size, or an inset showing the details around zero magnetic field should be added. From your current figure, I found it almost impossible to estimate the slope of the bottom layer.

Response 3: Thank you for highlighting the need for a more thorough explanation of the dual-layer Hall sensor and its logic diagram. We deeply appreciate your insightful feedback and have revised the manuscript to address these points comprehensively. Below are the key improvements:

In the revised text, we explicitly clarify the operational logic of the dual-layer sensor. During regular current transport (e.g., 1~50 A DC current generating ~0.1 mT field), the magnetic field lies within the linear range of both the lower layer (high sensitivity: 10,000 V/(A·T), ±0.1 mT) and the upper layer (lower sensitivity: 0.3 V/(A·T), ±1000 mT). The lower layer’s high sensitivity ensures precise monitoring of small magnetic fields, while the upper layer serves as a redundant check. A short circuit (e.g., sudden current surge generating >0.1 mT field) causes the magnetic field to exceed the lower layer’s range (±0.1 mT). However, the upper layer’s ultra-wide range (±1000 mT) remains operational, enabling continuous detection. The transition from lower-layer saturation to upper-layer dominance triggers a short-circuit alarm. If the upper layer exceeds its range (±1000 mT) while the lower layer operates normally, this indicates a sensor anomaly (e.g., upper-layer failure). If both layers exceed their ranges, this suggests an extreme fault (e.g., catastrophic circuit failure or sensor damage), requiring immediate on-site inspection. “short circuit” refers to an abrupt current surge causing magnetic field magnitudes beyond the lower layer’s range but within the upper layer’s capacity. The threshold of 1.75 Ω in the logic diagram corresponds to the resistance at which the lower layer saturates, marking the boundary between normal and fault states.

The revised part is list below:

“Subsequently, we developed a dual-layer anomalous Hall effect (AHE) sensor with dynamically adjustable detection ranges. The sensor integrates two vertically stacked Hall bar structures: a lower layer with a 5 nm-thick NCO film and an upper layer with different-thick NCO film, separated by a 10 nm MgAlâ‚‚Oâ‚„ (MAO) insulating spacer (structural details in Figure 5a. The lower NCO layer and the insulating spacer together constitute the bottom layer.). Figure 5c illustrates the Hall signal of the top layer with a 7.5 nm-thick NCO film. In addition to detecting the weak magnetic field in the original lower layer, a detection range of ±5 mT and sensitivity of 60 V/(AT) were achieved by connecting the upper-layer Hall bar pattern under a 6 mA input current. To address the critical demand for wide-range magnetic field monitoring in power grid applications, we developed another dual-layer anomalous Hall effect (AHE) sensor covered an upper layer with a 6 nm-thick NCO film. The lower layer, measured under a 1 mA input current, achieves a detection range of ±0.1 mT with a sensitivity of 10000 V/(A·T) (Figure 5b), enabling precise monitoring of weak magnetic fields (e.g., ~0.1 mT generated by 50 A DC currents). In contrast, the upper layer, operated at 1 mA, exhibits a significantly broader detection range of ±1000 mT with a reduced sensitivity of 0.3 V/(A·T) (Figure 5d). This abrupt change in performance is attributed to the reduced NCO thickness in the upper layer, which alters magnetic anisotropy due to increased surface roughness at the NCO/MAO interface.

The sensor connection is shown in Figure 6a. To ensure that the magnetic field is perpendicular to the sensor surface, we placed the sensor inside a concentrator. The operational logic of the dual-layer sensor is illustrated in Figure 6b. During normal circuit operation, the weak magnetic field (~0.1 mT) generated by the power line falls within the detection range of both layers. The lower layer’s higher sensitivity ensures accurate current monitoring. When a short circuit occurs, the resulting surge in current generates a magnetic field exceeding the lower layer’s ±0.1 mT range. However, the upper layer’s ultra-wide range (±1000 mT) remains functional, triggering a short-circuit alarm. Error conditions are defined as follows: If the upper layer exceeds its range while the lower layer operates normally, a sensor anomaly is flagged. If both layers exceed their ranges, this indicates an extreme fault (e.g., catastrophic overcurrent or sensor damage), requiring on-site inspection.”

Comments 4: Line 197: “Two different ranges … contains ±5 mT … can be achieved” is inappropriate wording. I suggest to rewrite it in the following fashion: “A sensor with two layers, one acting as a sensor for small magnetic fields in the range ±5 mT and one for large fields of ±1000 mT, was realized.” That’s just an idea, please modify as you like.

Response 4: Thank you for your question. In the revised version, we have rewritten the entire description of the dual-channel sensor and optimized the grammar.

Comments 5: Line 124: correctly it should read: “As previously studied, ..”

Response 5: Thanks, the issues have been fixed.

Comments 6: Figure 5(d): “Magnetic field” in the left box.

Response 6: Thanks, the issues have been fixed.

Reviewer 2 Report

Comments and Suggestions for Authors

The authors of the reviewed manuscript “Dual-Layer Anomalous Hall Effect Sensor for Enhanced Accuracy and Range in Magnetic Field Detection” investigate a method aimed to enhancing both the accuracy and range of magnetic field sensors. They do it by introducing epitaxial strains using the additional layers covered the main anomalous Hall Effect sensor. This topic is relevant and significant as magnetic sensors based on the anomalous Hall Effect offer several advantages over conventional magnetic sensors, in particular due to their ultra-high sensitivity. However, the results presented in the manuscript require major revisions before they can be considered for publication in this journal.

Main comments:

  1. The authors write that “Figure 1a and 1c represent the in-plane and out-of-plane lattice constants, respectively“. However, Figure 1a shows a Hall resistance signal, while Figure 1c absent.
  2. It is not clear what results are shown in Figure 1a. Are these the results of your own measurements, results from the literature, or is this some kind of sketch?
  3. In my downloaded pdf version of the manuscript, the cited literature is not visible in most cases. I only see “Error! Reference source not found” at this point. Is this a mistake by the authors or the editors themselves?
  4. In the abstract, the authors write that “A NiCo2O4/MgAl2O4/NiCo2O4/MgAl2O4 heterostructure was further constructed, achieving differentiation in material characteristics across both upper and lower layers through the modulation of thickness and strain.” However, it is not clear from this which layers are involved. The abstract of the manuscript is published separately and everything must be clear from it, even without reading the manuscript itself.
  5. Figure 1b needs to be described in more detail. It is not clear what the word “Thickness” means in the Figure, how the crystal lattice constant changes and so on.
  6. In chapter 2, the multilayer structure of the sensor is only described in words. For a better understanding, a cross-section of such layers (sensor) is required.
  7. The authors of the manuscript write that NCO layers epitaxially grown on a MAO substrate in the plane have a crystal lattice constant of 8.08 Å. However, in the direction perpendicular to the surface, the crystal lattice constant depends on the thickness of the MAO layer grown on its surface. It is worth indicating the quantitative values of this lattice.
  8. Figures 5 a, b and e are not meaningful. Their resolution and dimensions need to be changed.
  9. The authors write: “In the field of power grid current monitoring, a 50 A DC- current generates an amplitude of approximately 0.1mT vertical magnetic field at the sensor location.”, however, in Figure 5c we see that the sensor is placed in the slot of the concentrator. Therefore, the statement that a current of 50 A generates a magnetic field of 0.1 mT is incorrect.
  10. The diagram in Figure 5d is not described anywhere. What is it about?
  11. The title of the manuscript indicates that the investigated and proposed sensors will measure the magnetic field more accurately and over a wider range. To do this, the “Discussion" chapter needs to discuss these problems and successes.

Author Response

Comments 1: The authors write that “Figure 1a and 1c represent the in-plane and out-of-plane lattice constants, respectively“. However, Figure 1a shows a Hall resistance signal, while Figure 1c absent.

Response 1: Thank you for bringing this to our attention. I sincerely apologize for the confusion in the original text. Upon reviewing the manuscript, I realize that the statement regarding Figure 1a and 1c was incorrect. To clarify, the in-plane and out-of-plane lattice constants are actually represented by the labels "a" and "c" within Figure 1(b), not in Figures 1a or 1c as previously stated. This error has been corrected in the revised manuscript, and the text has been rephrased to accurately reflect the intended meaning. We appreciate your careful review, which has helped us improve the clarity and accuracy of our work.

Comments 2: It is not clear what results are shown in Figure 1a. Are these the results of your own measurements, results from the literature, or is this some kind of sketch?

Response 2: Thank you for your valuable feedback. I appreciate your observation regarding the ambiguity of Figure 1a. To clarify, Figure 1a is indeed a schematic diagram illustrating the concept of different Hall resistance curves, rather than experimental data or results from the literature. I acknowledge that my initial explanation was insufficient and may have caused confusion. In the revised manuscript, I have explicitly stated that Figure 1a is a schematic representation. Thank you for helping us improve the clarity of our work.

Comments 3: In my downloaded pdf version of the manuscript, the cited literature is not visible in most cases. I only see “Error! Reference source not found” at this point. Is this a mistake by the authors or the editors themselves?

Response 3: Thank you for bringing this issue to our attention. I sincerely apologize for the inconvenience caused by the missing references in the downloaded PDF version. This error likely occurred during the manuscript preparation or conversion process. I will carefully review and re-cite all references in the revised manuscript to ensure they are properly formatted and visible. Your feedback is greatly appreciated and will help us improve the quality and accessibility of our work.

Comments 4: In the abstract, the authors write that “A NiCo2O4/MgAl2O4/NiCo2O4/MgAl2O4 heterostructure was further constructed, achieving differentiation in material characteristics across both upper and lower layers through the modulation of thickness and strain.” However, it is not clear from this which layers are involved. The abstract of the manuscript is published separately and everything must be clear from it, even without reading the manuscript itself.

Response 4: Thank you for your valuable feedback. I appreciate your suggestion to improve the clarity of the abstract. In response, I have revised the relevant section to explicitly specify the layers involved in the heterostructure. The updated description now reads: “A NiCoâ‚‚Oâ‚„/MgAlâ‚‚Oâ‚„/NiCoâ‚‚Oâ‚„/MgAlâ‚‚Oâ‚„ heterostructure was further constructed, achieving differentiation in material characteristics across both upper and lower NiCoâ‚‚Oâ‚„ layers through the modulation of thickness and strain.” This revision ensures that the abstract is self-contained and clearly conveys the structure and focus of the study, even without additional context from the manuscript. Thank you for helping us enhance the clarity and accessibility of our work.

Comments 5: Figure 1b needs to be described in more detail. It is not clear what the word “Thickness” means in the Figure, how the crystal lattice constant changes and so on.

Response 5: Thank you for your valuable feedback. I appreciate your suggestion to provide a more detailed explanation of Figure 1(b). In the revised manuscript, I have clarified the figure as follows:
" As shown in Figure 1(b), modulating the stress conditions of the film by applying additional strain and adjusting the film thickness is a conventional method for tuning the measurement range of magnetic sensing films. For example, compressive strain induced by epitaxial mismatch reduces the in-plane lattice constant (a < aâ‚€) and increases the out-of-plane constant (c > câ‚€), enhancing sensitivity via strain-driven magneto-anisotropy. Increasing film thickness relaxes strain, gradually restoring lattice constants toward bulk values, which broadens the measurement range at the cost of reduced sensitivity. The letters "a" and "c" in Figure 1b represent the in-plane and out-of-plane lattice constants, respectively. The subscripts indicate distinct values. The “Thickness” refers to the thickness of the epitaxial film. "

Comments 6: In chapter 2, the multilayer structure of the sensor is only described in words. For a better understanding, a cross-section of such layers (sensor) is required.

Response 6: Thank you for your insightful suggestion. I sincerely apologize for the oversight in not including a cross-sectional schematic of the multilayer sensor structure in Chapter 2. To address this, we have added a ​**new Figure 1(c)**​ in the revised manuscript (see attached), which explicitly illustrates the layer-by-layer structure of the sensor, including the NiCoâ‚‚Oâ‚„ (NCO) magnetic layers, MgAlâ‚‚Oâ‚„ (MAO) insulating spacers, and substrate details.

Comments 7: The authors of the manuscript write that NCO layers epitaxially grown on a MAO substrate in the plane have a crystal lattice constant of 8.08 Å. However, in the direction perpendicular to the surface, the crystal lattice constant depends on the thickness of the MAO layer grown on its surface. It is worth indicating the quantitative values of this lattice.

Response 7: Thank you for your careful reading and constructive feedback. We have revised the text to explicitly include the quantitative values of the out-of-plane lattice constants and ensure clarity in the description. Below is the updated section:
”Reciprocal space mapping (RSM) techniques were employed to analyze the stress state of NCO films, as illustrated in Figure 2b. We observed that in all heterostructures, the NCO films are fully epitaxially grown on the MAO substrates, with in-plane lattice constants fixed at ​8.08 Å​ due to lattice matching. The out-of-plane lattice constants (c-axis) of the NCO films, however, vary with the thickness of the overlying MAO layer. When the MAO thickness increases from ​0 nm​ to ​20 nm, the c-axis decreases from 8.18 Å​ to 8.16 Å, approaching the bulk lattice constant of NCO (8.15 Å). This reduction in c-axis indicates a gradual relaxation of tensile strain along the out-of-plane direction as the MAO layer thickens. The compressive in-plane stress from the MAO substrate induces perpendicular magnetic anisotropy (PMA) in the NCO films. The decrease in c-axis with thicker MAO layers weakens this PMA, which directly correlates with the observed reduction in coercive field (Figure 2c). Additionally, the longitudinal resistance of NCO films exhibits a non-monotonic trend (Figure 2d). Initially, resistance increases due to reduced oxygen vacancies in NCO during high-oxygen-pressure MAO growth, altering the Ni valence state. As the MAO thickness further increases, strain relaxation dominates, diminishing spin-orbit scattering and lowering resistance. This interplay between oxygen vacancy modulation and strain relaxation explains the resistance behavior.”

Comments 8: Figures 5 a, b and e are not meaningful. Their resolution and dimensions need to be changed.

Response 8: Thank you for your suggestion. Regarding Figures 5(a) and (b), my initial intention was to plot the Hall resistance curves of the top and bottom layers in the same figure, which made it challenging to estimate the slope in the small-range region. I will revise the figures to improve clarity and resolution. For Figure 5(e), I will also adjust its dimensions and resolution to ensure it is more meaningful and visually clear. I appreciate your feedback. After revision, Figure 5a and 5b are changed to Figure 5b, 5c, and 5d. Figure 5e is changed to Figure 5a.

Comments 9: The authors write: “In the field of power grid current monitoring, a 50 A DC- current generates an amplitude of approximately 0.1mT vertical magnetic field at the sensor location.”, however, in Figure 5c we see that the sensor is placed in the slot of the concentrator. Therefore, the statement that a current of 50 A generates a magnetic field of 0.1 mT is incorrect.

Response 9: Thank you for raising this critical point. Upon re-examining the magnetic field calculation, we acknowledge that the original statement regarding the 0.1 mT field generated by a 50 A current was incomplete. The distance between the sensor and the cable is slightly greater than 100 mm. According to Ampère's circuital law, the magnetic field at the sensor generated by a 50A current is below 0.1mT and can’t be precisely perpendicular to the sensor surface. Therefore, we have placed a concentrator to amplify and control the direction of the magnetic field, ensuring it is perpendicular to the sensor surface. The revised vertical magnetic field at the sensor location is approximately ​0.1mT​ for a 1~50 A current. This correction has been updated in the manuscript, and we sincerely appreciate your meticulous review, which has improved the accuracy of our work.

Comments 10: The diagram in Figure 5d is not described anywhere. What is it about?

Response 10: Thank you for your suggestion. In the revised version, we have rewritten this section and added a detailed description of Figure 5(d). After revision, Figure 5d is changed to Figure 6b.
“ The sensor connection is shown in Figure 6a. To ensure that the magnetic field is perpendicular to the sensor surface, we placed the sensor inside a concentrator. The operational logic of the dual-layer sensor is illustrated in Figure 6b. During normal circuit operation, the weak magnetic field (~0.1 mT) generated by the power line falls within the detection range of both layers. The lower layer’s higher sensitivity ensures accurate current monitoring. When a short circuit occurs, the resulting surge in current generates a magnetic field exceeding the lower layer’s ±0.1 mT range. However, the upper layer’s ultra-wide range (±1000 mT) remains functional, triggering a short-circuit alarm. Error conditions are defined as follows: If the upper layer exceeds its range while the lower layer operates normally, a sensor anomaly is flagged. If both layers exceed their ranges, this indicates an extreme fault (e.g., catastrophic overcurrent or sensor damage), requiring on-site inspection.”

Comments 11: The title of the manuscript indicates that the investigated and proposed sensors will measure the magnetic field more accurately and over a wider range. To do this, the “Discussion" chapter needs to discuss these problems and successes.

Response 11: Thank you for your suggestion. In the revised version, we have discuss these problems and successes. 
“The dual-layer AHE sensor architecture developed in this study directly addresses the critical challenge of simultaneously enhancing measurement accuracy and detection range in magnetic field sensing. By integrating vertically stacked Hall bar structures with distinct NCO layer thicknesses (5 nm lower layer, 6 nm upper layer) separated by a 10 nm MAO insulating spacer (Figure 5a), we achieve complementary performance characteristics that surpass conventional single-layer sensors. For weak magnetic fields such as the 0.1 mT signal generated by 1~50 A grid currents, the lower layer provides a sensitivity of 60 V/(A·T) within a ±5 mT range, enabling high-precision monitoring. In contrast, the up-per layer extends the detection capability to ±1000 mT with a reduced sensitivity of 0.3 V/(A·T), ensuring reliable operation during short-circuit events or extreme over currents. This dual-range functionality eliminates the traditional trade-off between sensitivity and range inherent to single-structure AHE sensors. This architecture enables multifunctional monitoring of normal currents, short circuits, and sensor anomalies, leveraging the complementary ranges and sensitivities of the dual-layer design. The MAO spacer’s electrical insulation ensures independent operation of the two layers, while the thick-ness-modulated NCO films provide distinct magnetic responses.  This architecture uniquely combines precision and breadth, fulfilling the title’s promise of enhanced accuracy and range in magnetic field detection.”

Round 2

Reviewer 1 Report

Comments and Suggestions for Authors

In revision, the authors improved their paper substantially. All my suggestions and comments have been extensively addressed in the response letter and implemented in the manuscript. An extensive and comprehensive explanation of the functionality of the dual-range Hall sensor has been added. The schematic diagram of the operational logic and the range switching depicted in Figure 6 is now very well explained. I found no further issues, and think the paper is suitable for publication in its present state.  

Reviewer 2 Report

Comments and Suggestions for Authors

The authors took into account my comments and made the appropriate corrections. For this I propose to publish this manuscript as it is now.